# Estimation of Hourly Land Surface Heat Fluxes over the Tibetan Plateau by the Combined Use of Geostationary and Polar Orbiting Satellites

Lei Zhong[1,2,3], Yaoming Ma[4,5,6], Zeyong Hu[5,7], Yunfei Fu[1], Yuanyuan Hu[1], Xian Wang[1], Meilin Cheng[1], Nan Ge[1]

[1]School of Earth and Space Sciences, University of Science and Technology of China, Hefei 230026, China

[2]CAS Center for Excellence in Comparative Planetology, Hefei 230026, China

[3]Jiangsu Collaborative Innovation Center for Climate Change, Nanjing 210023, China

[4]Key Laboratory of Tibetan Environment Changes and Land Surface Processes, Institute of Tibetan Plateau Research, the Chinese Academy of Sciences, Beijing 100101, China

[5]CAS Center for Excellence in Tibetan Plateau Earth Sciences, Beijing 100101, China

[6] University of Chinese Academy of Sciences, Beijing 100049, China

[7] Northwest Institute of Eco-Envrionment and Resources, the Chinese Academy of Sciences, Lanzhou 730000, China

*Correspondence to:* Lei Zhong (zhonglei@ustc.edu.cn)

**Abstract.** Estimation of land surface heat fluxes is important for energy and water cycle studies, especially on the Tibetan Plateau (TP), where the topography is unique and the land-atmosphere interactions are strong. The land surface heating conditions also directly influence the movement of atmospheric circulation. However, high temporal resolution information on the plateau-scale land surface heat fluxes has lacked for a long time, which significantly limits the understanding of diurnal variations in land-atmosphere interactions. Based on geostationary and polar orbiting satellite data, the surface energy balance system (SEBS) was used in this paper to derive hourly land surface heat fluxes at a spatial resolution of 10 km. Six stations scattered throughout the TP and equipped for flux tower measurements were used to perform a cross-validation. The results showed good agreement between the derived fluxes and in situ measurements through 3738 validation samples. The root mean square errors (RMSEs) for net radiation flux, sensible heat flux, latent heat flux and soil heat flux were 76.63 Wm$^{-2}$, 60.29 Wm$^{-2}$, 71.03 Wm$^{-2}$ and 37.5 Wm$^{-2}$, respectively. The derived results were also found to be superior to the Global Land Data Assimilation System (GLDAS) flux products (with RMSEs for the surface energy balance components of 114.32 Wm$^{-2}$, 67.77 Wm$^{-2}$, 75.6 Wm$^{-2}$ and 40.05 Wm$^{-2}$, respectively). The diurnal and seasonal cycles of the land surface energy balance components were clearly identified, and their spatial distribution was found to be consistent with the heterogeneous land

surface conditions and the general hydrometeorological conditions of the TP.

## 1. Introduction

Mass and energy exchanges are constantly carried out between the land surface and the atmosphere above. At the same time, the weather, climate and environmental changes at multiple spatiotemporal scales are greatly influenced by such land-atmosphere exchanges. Land-atmosphere interaction is a popular topic not only in the field of atmospheric research but also in hydrology, geography, ecology and environmental sciences (Ye and Fu 1994). The impacts of land-atmosphere interactions on weather and climate change have been assessed through surface sensible heat flux, latent heat flux and momentum flux (Seneviratne et al, 2008; Ma et al, 2017). Developing a method to accurately derive surface heat fluxes has always been a primary focus in atmospheric science research.

The Tibetan Plateau (TP), with an average elevation of more than 4000 m, is also called 'the Third Pole' and 'the World Roof'. The thermal and dynamic effects caused by the TP's high elevation and relief have profound impacts on atmospheric circulation, the Asian monsoon and global climate change (Ye and Gao1979; Ma et al, 2006; Ma et al, 2008; Zhong et al, 2011; Zou et al, 2017; Zou et al, 2018). The interactions between TP multispheres, such as the atmosphere, hydrosphere, lithosphere, biosphere, and cryosphere, are the drivers of all these changes. The TP is also one of the most sensitive regions in response to global climate change (Liu et al, 2000). In recent years, some studies have argued that the major factor impacting the South Asian monsoon is the insulating effect of the southern mountain edges of the TP, rather than the elevated heating by the TP (Boos and Kuang 2010; Boos and Kuang 2013). However, some other studies have proven that the thermal effects of the TP are the main driving force of the South Asian summer monsoon (Wu et al, 2012; Wu et al, 2015). Obviously, opinions differ in understanding the thermal forcing by the TP. One of the most important reasons is that high spatial and temporal resolution data on land-atmosphere interactions, which can be used in different climate models, are still lacking. To study the characteristics of land-atmosphere interactions in the TP, it is necessary to estimate the surface energy heat fluxes with a fine spatial and temporal resolution over the TP.

Traditional surface energy flux measurements are not only expensive but also limited at the point scale and it is impossible to meet the need for a larger spatial scale with the complex terrain and landscapes of the TP. However, remote sensing provides the possibility of deriving surface heat fluxes at a regional scale (Ma et al, 2002; Zhong et al, 2014). The methods of estimating surface energy flux

by remote sensing can be roughly divided into three categories: the empirical (semiempirical) model, theoretical model and data assimilation system. The empirical (semiempirical) model is mainly based on an empirical formula between surface energy fluxes and surface characteristic parameters. The method itself is simple, but its applicability is limited. The basis of the theoretical model is the surface energy balance equation. The physical model mainly includes a single source model and a double source model. The single source model does not distinguish vegetation transpiration and soil evaporation but tends to consider them as a whole (Su, 2002; Jia et al, 2003; Roerink et al, 2000; Bastiaanssen et al, 1998; Allen et al, 2007). The double source model separates the vegetation canopy from the soil and calculates the soil temperature and canopy temperature. Then, the sensible heat flux and latent heat flux are calculated (Norman et al, 1995; Sánchez et al,2008). In recent years, land surface temperature (LST) and vegetation index data retrieved from satellites have been successfully assimilated in the variational data assimilation (VDA) frameworks to estimate surface heat fluxes (Crow and Kustas 2005; Bateni et al., 2013; Xu et al., 2014; Abdolghafoorian et al., 2017; Xu et al., 2019). This kind of method does not require any empirical or site-specific relationships and can provide temporally continuous surface heat flux estimates from discrete spaceborne LST observations (Xu et al., 2014).

Some studies have been carried out to estimate surface energy fluxes over the TP based on polar orbiting satellite data. Ma et al. (2003) estimated the surface energy flux of the CAMP (CEOP Asia–Australia monsoon project)/Tibet area using NOAA-14/AVHRR data. The results show that the estimated surface energy flux is in good agreement with the in situ measurements. Oku et al. (2007) used LST derived from the Geostationary Meteorological Satellite (GMS)-5 and other essential parameters from NOAA-AVHRR, ERA-40 to estimate land surface heat fluxes for regions above 4000 m over the TP. However, the coarse resolution of EAR-40 (25 km) and large error of LST (more than 10 K) introduced large uncertainties into the final results. Ma et al. (2009) estimated the surface characteristic parameters and surface energy flux of the northern TP in summer, winter and spring using a parameterized scheme for ASTER satellite data. Chen et al. (2013a) used observations from 4 sites in the TP to evaluate the results of the surface energy balance system (SEBS) model and optimize the thermodynamic roughness parameterization scheme for the underlying surface of bare soil. Based on Landsat TM/ETM+ data, Chen et al. (2013b) derived the surface energy flux of the Mount Everest area by using the enhanced SEBS model (TESEBS), which takes into account the influence of terrain

factors on solar radiation, and the SEBS model. The results show that the estimated results from the TESEBS model are superior to those from the SEBS model for high resolution satellite images.

At present, the estimation of the surface energy flux of the TP is mainly based on polar orbit satellite data. Because of the low temporal resolution of the polar orbit satellites, time series of land-atmosphere energy and water exchange data with high temporal resolution in the TP have not been retrieved to date, and the effective basic parameters for the climate model cannot yet be provided. In addition, one of the basic characteristics of the atmospheric boundary layer is its diurnal variation, and information on daily variations in surface energy flux is also lacking over the TP.

This paper mainly focused on how to acquire time series of energy flux data with high temporal resolution using a combination of geostationary and polar orbiting satellite data. First, the surface energy fluxes over the TP were estimated using the SEBS model with inputs from high temporal resolution LST from FY-2C data and other land surface characteristic parameters from polar orbiting satellite data. Then, the derived land surface heat fluxes were validated by flux tower measurements and were also compared with Global Land Data Assimilation System (GLDAS) flux products. The study area and datasets used in this study are introduced in section 2. The model description is given in section 3, followed by the results and discussion in section 4. The main conclusions are drawn in section 5.

## 2. Study Area and Data

The TP, located in southwest China, has an area of approximately $2.5 \times 10^6$ km$^2$ (Fig. 1) and is the largest plateau in China. With an average elevation of approximately 4000 m, the TP is also the highest plateau in the world, and the high elevation can directly influence the middle and upper layers of the atmosphere. Due to the harsh climate conditions and complex topography of the TP, the meteorological stations in this area are not only sparse but also unevenly distributed. A total of 6 meteorological stations are used for comparison with model estimates. Although these 6 stations are not scattered throughout the entire TP, they include several major land cover types (Zhong et al, 2010), and their elevation varies from 3000 m to 5000 m (Table 1). These stations are the only stations currently available, and each station is equipped to make four-component radiation measurements, soil moisture and temperature measurements, eddy-covariance measurements and conventional observation items such as wind speed (u), air temperature ($T_a$), specific humidity (SH) and air pressure ($P_s$).

Both the geostationary satellite Feng Yun 2C (FY-2C) and the polar orbiting satellite SPOT are used to retrieve the essential land surface characteristic parameters. The stretched visible and infrared spin scan radiometer (SVISSR) onboard FY-2C is used to derive the hourly LST with a spatial resolution of 5 km following the algorithms developed by our group (Hu et al, 2018). We should point out here that SVISSR has no infrared channel, which would be needed to derive normalized difference vegetation index (NDVI), albedo and emissivity. Suppose that these parameters (NDVI, albedo and emissivity) have little variation during a day, then, the product of the orbiting satellite SPOT is used instead. The spatial resolution for NDVI, albedo and emissivity is 1 km with a daily temporal resolution. All the above satellite data with a higher spatial resolution were resampled to match the resolution of the meteorological forcing data (Zou et al, 2018). The time period for all meteorological data and satellite data covers the whole year of 2008.

A forcing dataset developed by the Institute of Tibetan Plateau Research, Chinese Academy of Sciences (ITPCAS), is used as the model input in this study. The dataset has merged the observations from 740 operational stations of the China Meteorological Administration (CMA) with the corresponding Princeton global meteorological forcing dataset (He, 2010; Yang et al, 2010). The parameters used in this study are downward shortwave radiation ($R_{swd}$), downward longwave radiation ($R_{lwd}$), wind speed, air temperature, specific humidity and air pressure. All these parameters have a spatial resolution of 10 km and a temporal resolution of 3 hours (Table 2). A linear statistical downscaling method was used to derive hourly meteorological forcing data based on original 3-hour forcing data and in situ measurements in this study. The general idea is to establish an empirical relationship between each 3-hour in situ measurement. Then this relationship is applied to the meteorological forcing data.

The GLDAS products are produced by combining satellite and ground-based observations using advanced land surface modeling and data assimilation techniques (Rodell et al, 2004; Zhong et al, 2011). These products have been proved to simulate optimal fields of land surface states and fluxes in near-real time (Rodell et al, 2004). Here, 3-hour land surface heat flux products with a spatial resolution of 25 km are selected for comparison with satellite estimates.

Since 6 stations in Table 1 were not used in the ITPCAS meteorological forcing data, they can be used as independent data to validate the accuracy of the forcing meteorological data. The root mean square error (RMSE), mean bias (MB), mean absolute error (MAE) and correlation coefficient (R) are used to make a comparison between the ITPCAS forcing data and in situ meteorological data.

$$RMSE = \sqrt{\frac{\sum_{i=1}^{N}(x_i - obs_i)^2}{N}} \tag{1}$$

$$MB = \frac{\sum_{i=1}^{N}(x_i - obs_i)}{N} \tag{2}$$

$$MAE = \frac{\sum_{i=1}^{N}|x_i - obs_i|}{N} \tag{3}$$

$$R = \frac{\sum_{i=1}^{N}(x_i - \bar{x})(obs_i - \overline{obs})}{\sqrt{\sum_{i=1}^{N}(x_i - \bar{x})^2}\sqrt{\sum_{i=1}^{N}(obs_i - \overline{obs})^2}} \tag{4}$$

where $x_i$ and $obs_i$ are the estimation and measurement, respectively. $\bar{x}$ and $\overline{obs}$ are the average values of the estimation and measurement, respectively. As shown in Table 3, all six parameters show reasonable accuracy with the in situ measurements, which means these forcing parameters can be used as model inputs.

## 3. Model Description

Fig. 2 shows the general steps to derive the land surface heat fluxes in this paper and the SEBS model is used in this study. Because the surface energy balance has the four components of radiation ($R_n$), sensible heat flux ($H_s$), latent heat flux (LE) and soil heat flux ($G_0$), the energy balance equation can be written as

$$R_n = H_s + LE + G_0 \tag{5}$$

where $R_n$ can be determined by the surface radiation equation as

$$R_n = R_{swd}(1 - \alpha) + \varepsilon_a \sigma T_a^4 - \varepsilon_s \sigma T_s^4 \tag{6}$$

where $R_{swd}$ is the downwelling solar radiation at the land surface. Because there are no infrared channels on board FY-2C, NDVI, $\alpha$ and $\varepsilon_s$ are derived from SPOT/VGT data. $\alpha$ is the broadband albedo, which can be derived from the narrowband reflectance of VGT $\alpha_1, \alpha_2, \alpha_3 and \alpha_4$ (Zou et al, 2018). $\alpha_1, \alpha_2, \alpha_3 and \alpha_4$ refer to the reflectance of the blue band, red band, near infrared band and short wave infrared band, respectively.

$$\alpha = -0.8141\alpha_1 + 0.4254\alpha_2 + 1.2605\alpha_3 - 0.2902\alpha_4 + 0.1819 \tag{7}$$

$\sigma$ in equation (6) is the Stefan-Boltzmann constant ($5.76 \times 10^{-8}$ W·m$^{-2}$·K$^{-4}$). $\varepsilon_a$ and $\varepsilon_s$ are the emissivities of surface air and the land surface, respectively. $T_a$ and $T_s$ are the surface air temperature and LST, respectively. The hourly $T_s$ is derived from split window algorithms (Hu et al, 2018) based on two thermal bands of FY-2C.

The soil heat flux is determined by net radiation flux and vegetation coverage.

$$G_0 = R_n[\Gamma_c + (1 - f_c)(\Gamma_s - \Gamma_c)] \tag{8}$$

where $\Gamma_s$ and $\Gamma_c$ are ratios of soil heat flux and net radiation flux for bare soil and full vegetation cover, respectively. $f_c$ is vegetation coverage and can be derived from NDVI as follows.

$$f_c = \left(\frac{NDVI - NDVI_{min}}{NDVI_{max} - NDVI_{min}}\right)^2 \tag{9}$$

$$NDVI = \frac{\alpha_3 - \alpha_2}{\alpha_3 + \alpha_2} \tag{10}$$

By using the wind speed and air temperature at the reference height, the sensible heat flux, together with the friction velocity and Obukhov stability length, can be derived by solving the following nonlinear equations (11-13). Then, the latent heat flux can be estimated by applying equation (5).

$$L = -\frac{\rho \cdot c_p \cdot \theta_v \cdot u_*^3}{k \cdot g \cdot H_s} \tag{11}$$

$$u_* = k \cdot u \cdot \left[ln\left(\frac{z - d_0}{z_{0m}}\right) - \Psi_m\left(\frac{z - d_0}{L}\right) + \Psi_m\left(\frac{z_{0m}}{L}\right)\right]^{-1} \tag{12}$$

$$H_s = k \cdot u_* \cdot \rho \cdot c_p \cdot (\theta_0 - \theta_a) \cdot \left[ln\left(\frac{z - d_0}{z_{0h}}\right) - \Psi_h\left(\frac{z - d_0}{L}\right) + \Psi_h\left(\frac{z_{0h}}{L}\right)\right]^{-1} \tag{13}$$

where L is the Obukhov length, $c_p$ is the specific heat at constant pressure, $\theta_v$ is the surface potential virtual air temperature, $u_*$ is the friction velocity, $k = 0.4$ is the von Karman constant, g is the acceleration due to gravity, $H_s$ is the sensible heat flux, $u$ is the mean wind speed at reference height z, $d_0$ is the zero plane displacement height, $z_{0m}$ is the roughness height for momentum transfer, $z_{0h}$ is the roughness height for heat transfer, $\Psi_m$ is the stability correction function for momentum heat transfer, $\Psi_h$ is the stability correction function for sensible heat transfer and $\theta_0$ and $\theta_a$ are the potential temperatures at the surface and reference height, respectively.

## 4. Results and Discussion

### 4.1 Validation Against In Situ Flux Tower Measurements

With the aid of SPOT/VGT and FY-2C/SVISSR data, the surface energy budget components have been estimated using the SEBS model. The accuracy of these estimates needs to be validated before further analyses. A total of 6 stations over the TP equipped with eddy-covariance measurements were selected for validation (Table 1). These validation stations cover a variety of climates, land cover types and elevations. The in situ flux data have been flagged by steady state tests and developed conditions tests according to Foken and Wichura (1996) and Foken et al. (2004). Steady conditions mean that all

statistical parameters do not vary with time. The flux-variance similarity was used to test the development of turbulent conditions. A data quality of only QA<5 was chosen to make the comparison. As shown in Fig. 3a, 3b, 3c and 3d, the estimates of surface energy budget components show reasonable agreement with the in situ measurements. The RMSEs for the net radiation flux, sensible heat flux, latent heat flux and soil heat flux are 76.63 $Wm^{-2}$, 60.29 $Wm^{-2}$, 71.03 $Wm^{-2}$ and 37.5 $Wm^{-2}$, respectively. The total validation numbers (N) are more than 3837 to make the results much more representative and convincing. It should be noted that some bias exists between the estimated soil heat flux and ground measurements because soil heat flux is parameterized with net radiation flux (equation (8)). However, soil heat flux and net radiation flux do not have the same diurnal variation shape. The soil heat flux peak values are usually later than the net radiation flux peak values, which was not taken into account in the parameterization. Thus, development of a better parameterization scheme for soil heat flux is needed.

The high-quality, global land surface fields provided by GLDAS support weather and climate prediction, water resources applications, and water cycle investigations. Since the GLDAS data have been widely used, it is meaningful to compare our satellite estimations with these high-quality data to further prove the accuracy of our estimations. To test the robustness of our results, the surface energy budgets obtained from the GLDAS data are selected for comparison with the FY-2C estimations. The comparison shows that the accuracies of the surface energy budgets from the satellite estimation are much higher than those of the GLDAS products (Table 4). The RMSE of the net radiation flux is reduced from 114.32 $Wm^{-2}$ to 76.63 $Wm^{-2}$, while the values for sensible heat flux, latent heat flux and soil heat flux are reduced from 67.77 $Wm^{-2}$, 75.6 $Wm^{-2}$ and 40.05 $Wm^{-2}$ to 60.29 $Wm^{-2}$, 71.03 $Wm^{-2}$ and 37.5 $Wm^{-2}$, respectively. Therefore, the new energy budget products not only have a finer spatial (10 km) and temporal resolution (hourly) than traditional polar orbiting satellite retrievals (e.g. Ma et al. 2006; Ma et al. 2014; Zou et al. 2018) but also possess much higher accuracy than the data assimilation results from GLDAS. Although the SEBS algorithm was used in this study and in Oku et al. (2007) (Oku 07 hereinafter), the methods for deriving the land surface characteristic parameters, such as LST and albedo, are different (Hu et al., 2018; Oku and Ishikawa 2004; Zou et al., 2018). The higher accuracy and finer spatiotemporal resolution of input forcing data (10km, 3 hour) and land surface characteristic parameters derived from satellites make our results more superior than those of Oku 07. It should also be noted that there is only one station used to perform the validation in Oku 07, while six

stations with four major land cover types were used in this study to make the results much more robust. Moreover, our results cover the entire TP, while Oku 07 only cover the region above 4000 m in the TP.

However, some discrepancies for this new product should be pointed out here, which means improvements are still needed for the current products. The error sources may come from multiple aspects, such as the uncertainties of input forcing data, the accuracy of land surface parameters from satellite retrievals and some assumptions and simplification in the SEBS model itself. As shown in Fig. 4, three sites located in the northern, western and southeastern parts of the TP were randomly selected to perform the sensitivity analysis. All input meteorological forcing parameters in Table 3 ($R_{swd}, R_{lwd}$ , u, $T_a$, SH, $P_s$) are selected. The original sensible heat flux and latent heat flux from the SEBS model are used as reference values. The RMSEs of different forcing data are used as perturbations. As shown in Table 5, the sensible heat flux is highly sensitive to $R_{swd}$, u and $T_a$, while the latent heat flux is very sensitive to $R_{swd}$, $R_{lwd}$ and $T_a$. Both sensible heat flux and latent heat flux are not sensitive to errors of SH and $P_s$. As $R_{swd}$ varies from –68.5 Wm$^{-2}$ to 68.5 Wm$^{-2}$, the induced latent heat flux uncertainty ranges from -29.75 Wm$^{-2}$ to 35.86 Wm$^{-2}$. Similarly, the sensible heat flux is very sensitive to $T_a$. When $T_a$ has an uncertainty from -2.08 K to 2.08 K, the induced sensible heat flux uncertainty ranges from 14.64 Wm$^{-2}$ to -16.94 Wm$^{-2}$. Furthermore, the mismatch between in situ measurements at the point level and the scales at the pixel level or grid level may cause some errors. The scale problem is an important issue and should be accounted for. However, this issue goes beyond the scope of this study.

## 4.2  Multitemporal and Spatial Distribution of Surface Energy Budget Components

One-year observation data and satellite estimations at BJ station were selected for comparison. As shown in Fig. 5, the satellite results can reproduce both the diurnal and seasonal surface flux variations very well. At the daily temporal scale, all the surface heat fluxes increase with sunrise and reach their maximum at mid-day before decreasing again with sunset. A unique characteristic of the atmospheric boundary layer is its well-known diurnal variations. The diurnal pattern of derived surface heat fluxes is in agreement with the diurnal evolution of the surface atmospheric boundary layer because the surface energy budgets provide a driving force for the surface atmospheric boundary layer. Fig. 5 also shows that the flux values are usually positive during the day and become negative during the night. This feature means that the dynamic and thermal contrasts of land and atmosphere are totally different between day and night. The surface heat fluxes during the day are much larger than those during the

night. At the seasonal scale, the diurnal mean net radiation flux usually increases from January (15.88 $W \cdot m^{-2}$) to its maximum in June (129.93 $W \cdot m^{-2}$). Then, it decreases again from June to December (2.07 $W \cdot m^{-2}$). The variation trends for sensible heat flux and latent heat flux are quite opposite. Because the TP is greatly influenced by the Asian monsoon system and the vegetation intensity usually increases from May to September (Zhong et al, 2010), the sensible heat flux usually decreases while the latent heat flux usually increases from the premonsoon season to the monsoon season. However, from the monsoon season to the postmonsoon season, the sensible heat flux increases while the latent heat flux decreases. The largest daily average intensity of sensible heat flux was found in April (34.97 $W \cdot m^{-2}$) while that for latent heat flux was found in June (69.09 $W \cdot m^{-2}$). As shown by the surface radiation balance equation (equation (6)), the downward shortwave radiation is the main incoming energy. A comparison was made between the forcing data and in situ downward radiation at BJ station. From June to August, the monthly diurnal MB was -4.87 $Wm^{-2}$, which explains why derived net radiation flux was underestimated by the SEBS model from June to August. This phenomenon was also found in the study by Yang et al. (2010). As for the time period from January to May, the underestimation of sensible heat flux was mainly caused by the negative bias of the land-atmosphere air temperature difference. The MB for the land-atmosphere difference could be -5.69 K from January to May. As there is a complementary relationship between sensible heat flux and latent heat flux, the corresponding latent heat flux tends to be overestimated.

A clear diurnal variation in hourly sensible heat flux and latent heat flux maps over the entire TP is shown in Fig. 6. Similar to the diurnal variations in net radiation flux, the amplitude of the sensible heat flux is relatively small before sunrise. Then, the sensible heat flux increases quickly until it reaches its maximum at approximately 14:00 (local standard time). After this time, sensible heat flux decreases gradually and tends to be smooth at night. The spatial distribution of sensible heat flux is somewhat complicated. In general, because of the sparse vegetation coverage and limited soil moisture in the western TP, the sensible heat flux is much lower than that in other parts of the TP. The latent heat flux tends to be zero before sunrise. With more solar energy after sunrise and much more evaporation from the soil and transpiration of vegetation, the latent heat flux rises gradually and reaches its maximum at 14:00. The spatial distribution of latent heat flux correlates well with the land surface conditions. In the southeastern part of the TP, the climate conditions are warm and wet. Thus, the vegetation density is much higher than that in the northwestern part. From southeast to northwest, the vegetation changes

from forest, meadow, and grassland to sands and gravels, and the latent heat flux decreases accordingly.

## 5. Conclusions and Remarks

A typical characteristic of the atmospheric boundary layer is diurnal variation. Limited information has been acquired to understand plateau-scale land-atmosphere interactions, especially their energy and water transfers, because of the limitation of point-scale observation and the low temporal resolution of polar orbiting satellites. In this study, polar orbiting satellite data were used to retrieve land surface characteristic parameters such as NDVI, vegetation coverage, albedo and emissivity. These parameters can be considered to have relatively very small diurnal variation but large seasonal variation. For other parameters with more typical diurnal variations, such as LST, the geostationary satellite FY-2C was used to retrieve plateau-scale LST. Other parameters with typical diurnal characteristics, such as downward longwave and shortwave radiation, air temperature, specific humidity, wind speed and air pressure, were derived from ITPCAS meteorological forcing data. Based on the SEBS model and the above inputs, a time series of hourly land surface heat flux data over the TP was derived. The new dataset has a fine spatial resolution of 10 km. According to the validation with 6 field stations (more than 3800 samples), the high correlation coefficients and low RMSEs indicate that the estimated land surface heat fluxes are in good agreement with the ground truth. Furthermore, the estimates were compared with the GLDAS flux data, which were thought to have high quality. The results showed that most derived variables were superior to the GLDAS data. Based on this new dataset, the diurnal cycle of land surface heat fluxes was clearly identified. Moreover, the seasonal variations were found to be influenced by the Asian monsoon system. This new dataset can help to understand and quantify the diurnal variations in the land surface heating field, which are very important for atmospheric circulation and weather changes in the TP, especially in winter and spring when the main heating source is from the land surface. This dataset can also help to evaluate the results of numerical models. The uncertainties of input forcing data, the accuracy of land surface parameters from satellite retrievals, the mismatch between different scales and some assumptions and simplification in the SEBS model itself lead to some discrepancies between the estimation and observation. Because of the relatively homogeneous land surface conditions of the field stations, the spatial scale mismatch between different data should have been minimized in our study. Scintillometry is possibly the most convenient method to measure fluxes at a scale of 1-10 km. Unfortunately, this device is lacking over

the TP. If we have enough in-situ measurements within a grid scale of 10 km or 25 km, an average or weighted average of measurements can be directly used to reduce some uncertainties caused by scale mismatch. However, for well-known reasons, it is very difficult to carry out such measurements in the TP with the harsh environment and climate conditions. For the next step, it is worthwhile to examine subpixel surface heat fluxes using techniques such as the temperature-sharpening method. Additionally, the FY-4 satellite with much higher spatial, temporal and spectral resolution will provide the opportunity to monitor land-atmosphere interactions in much more detail.

**Data availability.** The ground-based measurements used in this study were obtained from the Third Pole Environment Database (http://www.tpedatabase.cn/portal/MetaDataInfo.jsp?MetaDataId=43). The SPOT data can be downloaded from https://www.vito-eodata.be/PDF/portal/Application.html. The FY-2C data can be downloaded from http://satellite.nsmc.org.cn/portalsite/Data/DataView.aspx. The forcing data set for this study can be obtained from http://dam.itpcas.ac.cn/chs/rs/?q=data.

**Author contributions.** LZ designed the study and performed the SEBS model with help from YM and ZH. XW, MC and NG collected the analyzed the in-situ flux data and forcing data. YH and LZ retrieved the land surface parameters form FY-2C and SPOT data. LZ wrote the manuscript with help from YM and YF. All commented on the paper.

**Competing interests.** The authors declare that they have no conflict of interest.

**Acknowledgements.** This research was jointly funded by Strategic Priority Research Program of Chinese Academy of Sciences (Grant No. XDA20060101), the National Natural Science Foundation of China (Grant No. 41875031, 41522501, 41275028, 41661144043, 41830650), the Chinese Academy of Sciences (Grant No. QYZDJ-SSW-DQC019) and CLIMATE-TPE (ID 32070) in the framework of the ESA-MOST Dragon 4 Programme. We would like to thank the three anonymous reviewers for their valuable comments.

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

**Table 1: Ground measurement sites.**

| Sites | Longitude (°E) | Latitude (°N) | Elevation (m) | Land cover |
|---|---|---|---|---|
| BJ | 91.899 | 31.369 | 4509.0 | Plateau meadow |
| D105 | 91.943 | 33.064 | 5039.0 | Plateau grassland |
| MS3478 | 91.716 | 31.926 | 4620.0 | Plateau meadow |
| Linzhi | 94.738 | 29.765 | 3326.0 | Slope grassland |
| Nam Co | 90.989 | 30.775 | 4730.0 | Plateau grassland |
| QOMS | 86.946 | 28.358 | 4276.0 | Gravels |

**Table 2: Summary of the input datasets used for calculating land surface heat fluxes.**

| Variables | Data Source | Resolution | |
|---|---|---|---|
| | | **Spatial** | **Temporal** |
| $T_s$ | FY-2C/SVISSR | 5 km | hourly |
| NDVI | SPOT/VGT | 1 km | daily |
| $P_v$ | SPOT/VGT | 1 km | daily |
| $\alpha$ | SPOT/VGT | 1 km | daily |
| $\varepsilon_s$ | SPOT/VGT | 1 km | daily |
| $R_{swd}$ | ITPCAS | 10 km | 3-hour |
| $R_{lwd}$ | ITPCAS | 10 km | 3-hour |
| u | ITPCAS | 10 km | 3-hour |
| $T_a$ | ITPCAS | 10 km | 3-hour |
| SH | ITPCAS | 10 km | 3-hour |
| $P_s$ | ITPCAS | 10 km | 3-hour |

**Table 3: Validation of the forcing data.**

| Variables | RMSE | MB | MAE | R | N |
|---|---|---|---|---|---|
| $R_{swd}$ (W·m$^{-2}$) | 68.50 | -4.73 | 37.38 | 0.974 | 1048 |
| $R_{lwd}$ (W·m$^{-2}$) | 20.98 | -8.49 | 16.98 | 0.954 | 1048 |
| u (m·s$^{-1}$) | 1.71 | -1.01 | 1.28 | 0.793 | 1440 |
| $T_a$ (K) | 2.08 | -0.045 | 1.08 | 0.975 | 1440 |
| SH (kg·kg$^{-1}$) | $0.56\times10^{-3}$ | $-0.76\times10^{-4}$ | $0.37\times10^{-3}$ | 0.981 | 1438 |
| $P_s$ (hPa) | 8.51 | -2.25 | 6.53 | 0.865 | 1440 |

**Table 4: Comparison of derived flux data product and GLDAS against in situ measurements (Units: Wm$^{-2}$).**

| Model | Indicators | $R_n$ | $H_s$ | LE | $G_0$ | $R_{swu}$ | $R_{lwu}$ |
|---|---|---|---|---|---|---|---|
| SEBS | RMSE | 76.63 | 60.29 | 71.03 | 37.5 | 49.81 | 52.99 |
| | MB | -3.11 | -22.13 | 8.01 | 7.81 | 11.74 | -34.93 |
| | MAE | 50.49 | 45.67 | 48.79 | 28.43 | 26.88 | 39.31 |
| | R | 0.935 | 0.789 | 0.772 | 0.791 | 0.900 | 0.798 |
| | N | 4720 | 4554 | 3865 | 3837 | 4898 | 4721 |
| GLDAS | RMSE | 114.32 | 67.77 | 75.60 | 40.05 | 56.97 | 45.18 |
| | MB | 23.43 | 27.88 | -10.35 | -4.00 | -15.42 | -28.06 |
| | MAE | 81.90 | 47.48 | 44.89 | 30.52 | 31.35 | 31.61 |
| | R | 0.836 | 0.807 | 0.660 | 0.755 | 0.779 | 0.870 |
| | N | 1633 | 1580 | 1341 | 1329 | 1633 | 1633 |

5    **Table 5: Uncertainties for each meteorological forcing variable and the induced changes in $H_s$ and LE.**

| Variables | Assumed Uncertainty | Induced Uncertainty of $H_s$ | Induced Uncertainty of LE |
|---|---|---|---|
| $R_{swd}$ (W·m$^{-2}$) | -68.50~68.5 | -12.34~6.22 (-8.05%~4.06%) | -29.75~35.86 (-17.92%~21.60%) |
| $R_{lwd}$ (W·m$^{-2}$) | -20.98~20.98 | -2.50~2.50 (-1.63%~1.63%) | -15.54~15.54 (-9.36%~9.36%) |
| u (m·s$^{-1}$) | -1.71~1.71 | -9.47~7.31 (-6.18%~4.77%) | 9.47~-7.31 (5.71%~-4.41%) |
| $T_a$ (K) | -2.08~2.08 | 14.64~-16.94 (9.55%~-11.05%) | -14.64~16.94 (-8.82%~10.20%) |
| SH (kg·kg$^{-1}$) | $-0.56\times10^{-3}$~$0.56\times10^{-3}$ | -0.01~0.01 (-0.01%~0.01%) | 0.01~-0.01 (0.01%~-0.01%) |
| $P_s$ (hPa) | -8.51~8.51 | -0.01~0.01 (-0.01%~0.01%) | 0.01~-0.01 (0.01%~-0.01%) |

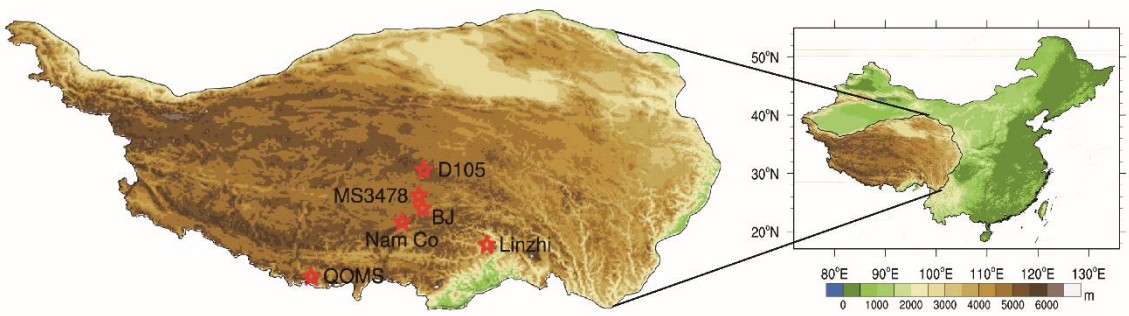

**Figure 1: Location of the Tibetan Plateau. The right panel illustrates the location of the Tibetan Plateau in China. The left panel shows the spatial distribution of eddy-covariance stations in the Tibetan Plateau. The pentagrams represent the eddy-covariance stations in the Tibetan Plateau. The legend of the color map indicates the elevation above mean sea level in meters.**

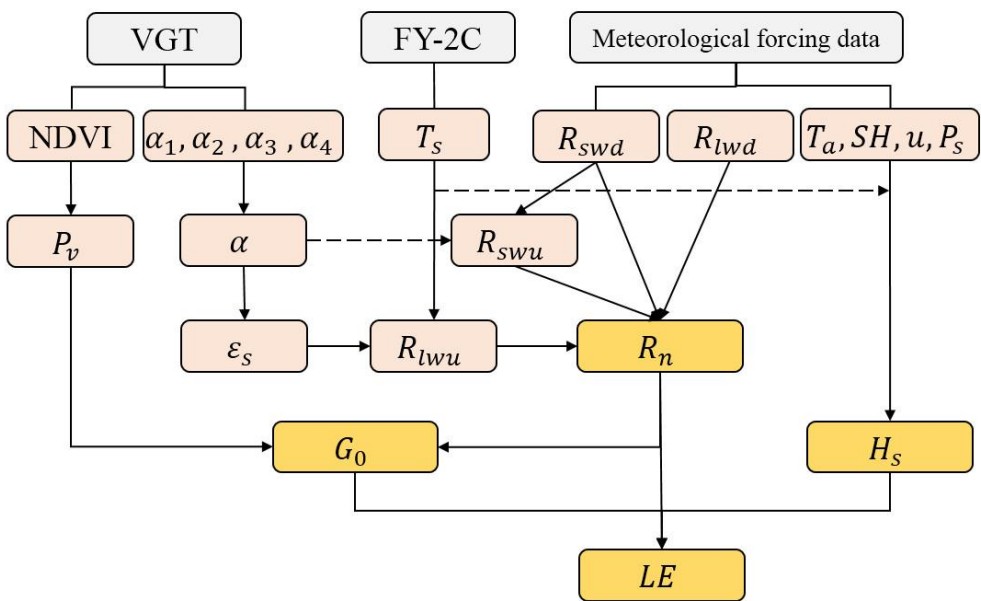

**Figure 2: Flowchart of the flux estimation method to determine the net radiation flux, sensible heat flux, latent heat flux by combining VGT, FY-2C and meteorological forcing data.**

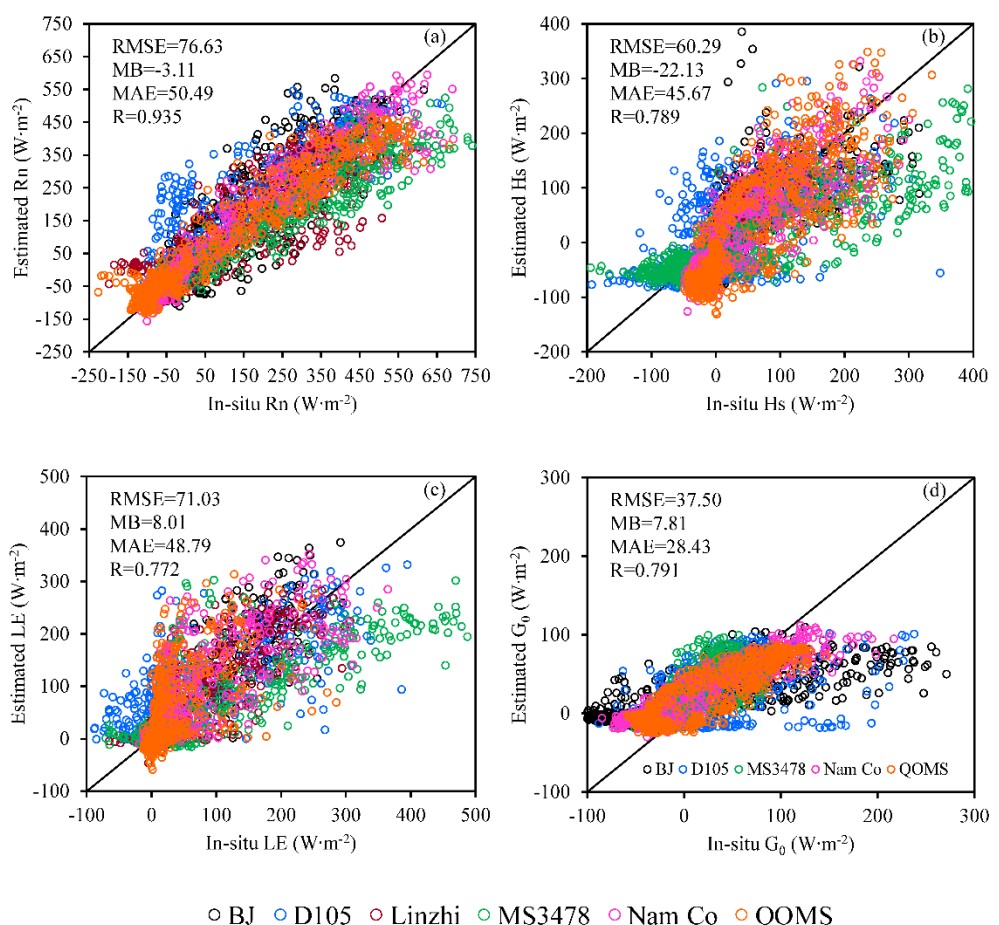

**Figure 3: Validation of surface heat fluxes estimated by the SEBS model with in situ measurements (a. Net radiation flux; b. Sensible heat flux; c. Latent heat flux; d. Soil heat flux). The legend with different colors indicates the six stations (BJ, D105, Linzhi, MS3478, Nam Co and QOMS) involved in the validation.**

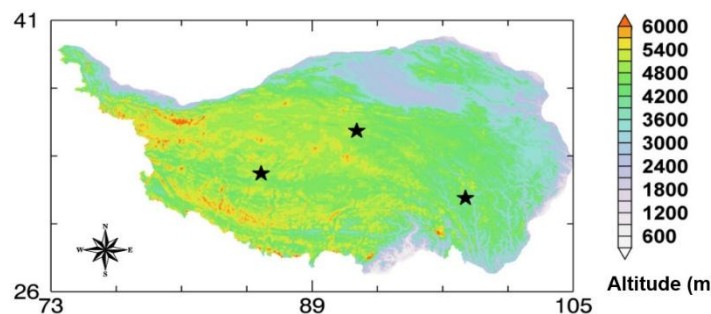

**Figure 4: Locations of the three sites (marked by pentagrams) used to carry out sensitivity tests of the meteorological forcing input data. The legend of the color map indicates the elevation above mean sea level in meters.**

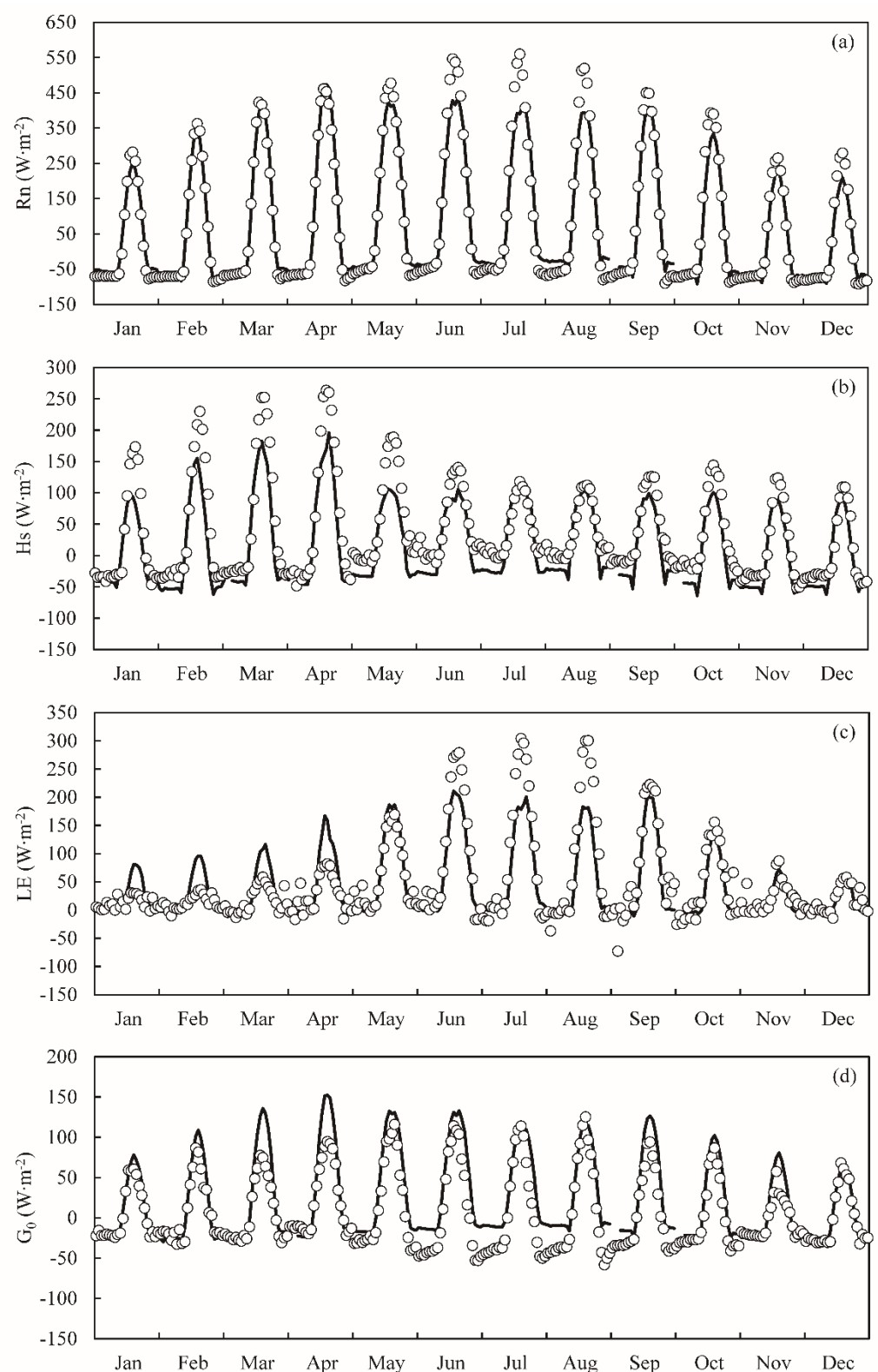

**Figure 5: Time series of monthly mean diurnal change in surface energy fluxes (units: Wm$^{-2}$) observed by in situ measurements (circle) and those estimated by using the SEBS model (curve) at the BJ station in 2008 (a. net radiation flux; b. sensible heat flux; c. latent heat flux; d. soil heat flux).**

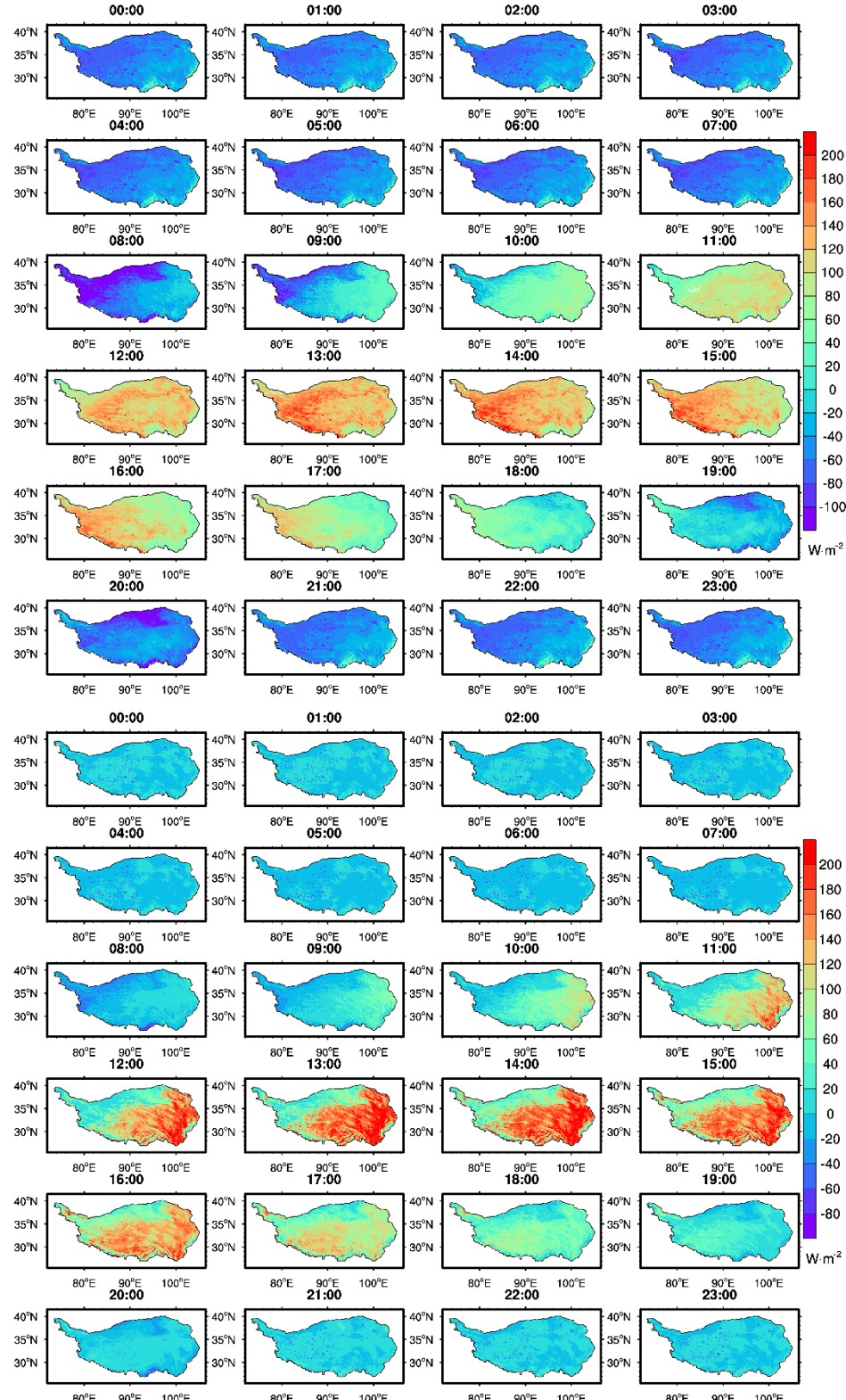

**Figure 6: Annual mean spatial distribution and diurnal cycle of sensible heat flux (top panels) and latent heat flux (bottom panels) in 2008 over the TP.**