# Peer review of "Estimation of Hourly Land Surface Heat Fluxes over the Tibetan Plateau by the Combined Use of Geostationary and Polar Orbiting Satellites"

_Atmospheric Chemistry and Physics, 2018_

## Referee Comment (RC1) · Anonymous Referee #2 · 6 Jan 2019

Comments to the Authors:

This is an integral work for estimation of land surface heat fluxes based on remote-sensing data, reanalysis meteorological data, and in-situ observations. The derived land surface heat flux, more like a heat flux dataset, was evaluated using observations of six eddy-covariance sites on the Tibetan Plateau (TP). And then, the diurnal and seasonal variations of the heat fluxes were also analyzed. This is of general interest for the readers of this journal.

The TP is notorious for its lack of meteorological observations, which cripples the predictive power of numerical models for this region. The land surface heat fluxes are

crucial for understanding energy and water cycle and also are the boundary conditions for numerical weather and climate simulations. This paper provides an integral investigation for land surface heat fluxes over the TP which will helps better understanding the land-atmosphere interactions over this region. More importantly, this paper is one of the very few works to estimate land surface heat fluxes over the TP using high temporal resolution geostationary satellite data.

The manuscript is well organized. Numbers of work are integrated into this paper, and abundant discussions are presented as well. I suggest acceptance after a minor revision.

Specific issues:

P1, L16: "which" → "where"

P1, L18-19: Change the sentence to "However, the high temporal-resolution information about the plateau-scale land surface heat fluxes has lacked for a long time, which significantly limit the understanding of diurnal variations in land-atmosphere interactions."

P1, L20: "a" → "the".

P1, L21: "with a spatial resolution" → "at a spatial resolution".

P4, L9: The sentence "These stations are the only stations currently available. . . . . ." is not accurate. I am quite sure that there are other eddy-covariance sites on the TP apart from the six stations mentioned in the paper.

P7, Equation (11) and (13): "Hs" → "Hs".

P8, L4: I do not think "Zhong et al., 2011" is a proper reference here. Perhaps you cite the paper which introduces the production of GLDAS data.

P8, L12: Provide some references for "traditional polar orbiting satellite" to strengthen your argument.

P10, L23: "land-atmosphere heat flux data" → "land surface heat flux data".

P10, L24: Delete "using a combination of geostationary and polar orbiting satellite data".

The English need substantial improvement. Please find a native speaker to help you to polish the manuscript.

---

## Referee Comment (RC2) · Anonymous Referee #4 · 16 Jan 2019

High temporal resolution surface heat fluxes are very important for land-atmosphere interactions. In this manuscript, land surface temperature from polar and geostationary satellite are both used and fed into surface energy balance equation. The results are validated with flux tower observations, and finally hourly surface heat fluxes with 5 km spatial resolution are generated over TP based on the developed SEB scheme. Generally, the manuscript is interesting and well written. It can be published with minor revisions.

Special comments:

- Page 2, Line 30: I think the authors missed an important kind of method (data assim-

ilation method) for surface heat flux estimations based on remotely sensed LST. Some reference are as follows,

Abdolghafoorian, A., Farhadi, L., Bateni, S.M., Margulis, S., Xu, T.R. (2017). Characterizing the effect of vegetation dynamics on the bulk heat transfer coefcient to improve variational estimation of surface turbulent fluxes. J. Hydrometeorol. 18, 321–333.

Bateni, S.M., Entekhabi, D., & Castelli, F. (2013), Mapping evaporation and estimation of surface control of evaporation using remotely sensed land surface temperature from a constellation of satellites, Water Resour. Res., 49, 950-968, doi:10.1002/wrcr.20071.

Crow, W.T., & Kustas, W.P. (2005). Utility of assimilating surface radiometric temperature observations for evaporative fraction and heat transfer coefficient retrieval, Bound-Lay. Meteorol., 115(1), 105-130, doi:10.1007/s10546-004-2121-0.

Xu, T.R., Bateni, Liang, S.M., Entekhabi, S.D., & Mao, K. (2014). Estimation of surface turbulent heat fluxes via variational assimilation of sequences of land surface temperatures from Geostationary Operational Environmental Satellites, J. Geophys. Res., 119, 10,780-10,798, doi:10.1002/2014JD021814.

Xu, T.R., He, X.L., Bateni, S.M., Auligne, T., Liu, S.M., Xu, Z.W., Zhou, J., Mao, K.B. (2019). Mapping Regional Turbulent Heat Fluxes via Variational Assimilation of Land Surface Temperature Data from Polar Orbiting Satellites, Remote Sensing of Environment, 221, 444-461, https://doi.org/10.1016/j.rse.2018.11.023.

- How to derive 5 km and hourly surface heat fluxes with 10 km and 3 hour forcing data?

- In equation 5, sensible heat flux is represented as Hs, while it is H in equation 11. They should be the same in one manuscript.

- What is the time period of this study? as well as validation results in Table 3.

- Figure2: the 'ITPCAS' is a name of institute, not data. It should be changed into

'Meteorological data' or something else.

- Figure 3: the estimated G0 has a big bias against ground measurements. This is because G0 is parameterized with Rn. G0 and Rn do not have the same diurnal variation shape. The G0 peak values are usually later than Rn. However, the parameterization did not consider this. The authors may discuss this in the manuscript.

- Figure 4: usually, the observations were drawn by open cycles, and estimations are drawn by solid lines.

- Why Rn is underestimated from June to Aug. at BJ site in figure 4? Why H (LE) is underestimated (overestimated) from Jan. to May? The authors should give some explanations.

- Figure 5: the authors give two days of diurnal cycles over TP. The results are from which day and which year? It should be noted on figure 5. In addition, why you choose these two days?

---

## Referee Comment (RC3) · Anonymous Referee #1 · 22 Jan 2019

Observations of land surface heat fluxes over the QTP are essential for understanding the land-atmosphere interactions. However, limited by the small amount of land-atmosphere monitoring stations and sparse spatial coverage, it is difficult to quantify the responses of the land-atmosphere interactions under the condition of climate warming on the QTP. This study aims to provide a plateau-scale product with a notable advantage of hourly-resolution using the SEB model in conjunction with the observations from polar and geostationary satellites. As we know that the temporal resolution of land surface heat fluxes is highly dependent on the forcing in various modelling approaches. In general, temperature and wind speed are two key input variables for the latent heat flux and the turbulent flux, respectively. The input variables in this study use the hourly temperature observations and other observations with a three-hour resolution. As a result, the reliability of the turbulent flux might be problematic when using the energy balance equation for calculation, and its accuracy is even worse than the 3-hourly product using data assimilation approach (e.g., GLDAS). A rigorous analysis of the accuracy is required to consolidate the proposed method. Given the present analysis, the current conclusion of hourly-resolution is not convincing for me. Considering other issues, a substantial revision is needed for this manuscript.

**Major issues:**

1. Since forcing data is lack of homogeneity in temporal and spatial resolution, the authors should discuss their impacts on the accuracy of the product. The authors declaimed a spatial resolution of 5 km, but it has been changed to 10 km in the new version (no rational explanation in the text). I think the authors should cope with the similar problem for the temporal resolution. As mentioned above, the methodology needs a rigorous analysis of the accuracy of the estimated land surface fluxes. Besides, I did not find the description of how to use the 3 hour-forcing in the SEBS model to produce the hourly product.

2. The major supporting for the conclusion of a better performance of the proposed product than the GLDAS produce is based on the comparison with the observational data. The authors use the Bowen ratio calibration method to improve the observed data. We know the validity of the Bowen ratio method varies distinctly in different environments due to the different fulfillment of assumptions. As a result, certain biases will be brought into the observational data, and this can mislead the comparison. First, it is not clear in the text that if the comparison is under the same condition that the observational data are all corrected with the Bowen ratio method. Second, even if using the similar observational data for the comparison, the biases from the correction can still distort the RMSE. Hence, I would suggest directly using the observed data for comparison. Besides, since the data quality of eddy covariance measurements may vary at the 6 stations, comparison on the indicators like RMSE at each station separately may provide more information.

3. The product provided by the authors is produced based on the input data with a spatial resolution no less than 10 km. The authors compare it with a product with a spatial resolution of 25 km. While the scale of the stations normally represents a scale of about less than 1 km. The authors should give some explanation about their comparability.

**Minor issues:**

1. P5-line 13-25: the authors validate the forcing data and find the notable variance. These differences can further propagate to the product. Please discuss its relation to the final product.

2. P8-line 2-6: The introduction of the GLDAS dataset should not belong to Result. The authors should introduce it more in light of its importance for comparison.

3. P8-line 5: what high accuracy?

4. P9-line 15-27: the authors describe the feature of diurnal variation of hourly flux map. Are there any special in comparison on our general understanding?

5. Table 4: add values of the same indicators for all sites.

6. Figure 1: the caption is too brief. The same problems for other plots. What is the right plot?

7. Figure 4: the scale of the axis is misleading. Besides, how do you choose the representative days for each month? Choose the nice one? Please describe what they are in subpanels.

---

## Author Response (AR1)

Dear Editor,

On behalf of my co-authors, I'm submitting our revised manuscript for possible publication in "Atmospheric Chemistry and Physics".

Thank you very much for your great efforts and high efficiency on evaluating our submission. We would also like to sincerely thank three anonymous reviewers for their constructive comments which are very helpful for us to improve our manuscript. We have carefully considered and fully addressed all comments. Below are the detailed point-by-point responses to the review comments. For clarity, the referees' comments are listed in black italics, while our responses and changes in the manuscript are shown in blue. We also mention where we make necessary changes in the revised manuscript by indicating page and line numbers in our responses. The marked manuscript was also uploaded to be easily reviewed.

We look forward to hearing from you soon.

Yours truly,
Lei Zhong et al.

**Response to Reviewer #1**

*Observations of land surface heat fluxes over the QTP are essential for understanding the land-atmosphere interactions. However, limited by the small amount of land-atmosphere monitoring stations and sparse spatial coverage, it is difficult to quantify the responses of the land-atmosphere interactions under the condition of climate warming on the QTP. This study aims to provide a plateau-scale product with a notable advantage of hourly-resolution using the SEB model in conjunction with the observations from polar and geostationary satellites. As we know that the temporal resolution of land surface heat fluxes is highly dependent on the forcing in various modelling approaches. In general, temperature and wind speed are two key input variables for the latent heat flux and the turbulent flux, respectively. The input variables in this study use the hourly temperature observations and other observations with a three-hour resolution. As a result, the reliability of the turbulent flux might be problematic when using the energy balance equation for calculation, and its accuracy is even worse than the 3-hourly product using data assimilation approach (e.g., GLDAS). A rigorous analysis of the accuracy is required to consolidate the proposed method. Given the present analysis, the current conclusion of hourly-resolution is not convincing for me. Considering other issues, a substantial revision is needed for this manuscript.*

**Author Response:** We would like to thank Reviewer #1 for the insightful and constructive comments. All your comments and suggestions are very helpful for improving our manuscript. We have carefully considered and addressed all of these comments, and significantly revised our manuscript. Please find our point-by-point response below.

**Major issues:**

*(1) Since forcing data is lack of homogeneity in temporal and spatial resolution, the authors should discuss their impacts on the accuracy of the product. The authors declaimed a spatial resolution of 5 km, but it has been changed to 10 km in the new version (no rational explanation in the text).I think the authors should cope with the similar problem for the temporal resolution. As mentioned above, the methodology needs a rigorous analysis of the accuracy of the estimated land surface fluxes. Besides, I did not find the description of how to use the 3 hour-forcing in the SEBS model to produce the hourly product.*

**Author Response:** Thank you for the above comments. The lack of homogeneity in the temporal and spatial resolution, mainly exists in the meteorological forcing data because its spatial and temporal resolution are lower than those of other satellite derived inputs. In addition to some remarks about this issue in Section 5 (P11, L27-30; P12, L1-4) , we also performed some sensitivity tests to verify how the RMSEs of forcing data can affect the sensible heat flux and latent heat flux. As shown in Figure 4, three sites located in the northern, western and southeastern part of the TP were randomly selected to perform the sensitivity analysis. All input meteorological forcing parameters in Table 3 ($R_{swd}, R_{lwd}$ , u, $T_a$, SH, $P_s$) are selected. The original sensible heat flux and latent heat flux from the SEBS model are used as reference values. The RMSEs of different forcing data were used as perturbations. As shown in Table 5, the sensible heat flux is highly sensitive to $R_{swd}$, u and $T_a$, while the latent heat flux is very sensitive to $R_{swd}$, $R_{lwd}$ and $T_a$. Both sensible heat flux and latent heat flux are not sensitive to errors of SH and $P_s$. As the $R_{swd}$ has a variation from –68.5 Wm$^{-2}$ to 68.5 Wm$^{-2}$, the induced latent heat flux uncertainty ranges from -29.75 Wm$^{-2}$ to 35.86 Wm$^{-2}$. Similarly, the sensible heat flux is very sensitive to $T_a$. When $T_a$ has an uncertainty from -2.08 K to 2.08 K, the induced sensible heat flux uncertainty ranges from 14.64 Wm$^{-2}$ to -16.94 Wm$^{-2}$. All the above works has been added to the revised manuscript. (P9, L6-16)

[Figure]

**Figure 4: Locations of the three sites (marked by pentagrams) used to carry out sensitivity tests of the meteorological forcing input data. The legend of the color map indicates the elevation above mean sea level in meters.**

**Table 5: Uncertainties for each meteorological forcing variable and the induced changes for H$_s$ and LE.**

| Variables | Assumed Uncertainty | Induced Uncertainty of H$_s$ | Induced Uncertainty of LE |
|---|---|---|---|
| $R_{swd}$ (W·m$^{-2}$) | -68.50~68.5 | -12.34~6.22 (-8.05%~4.06%) | -29.75~35.86 (-17.92%~21.60%) |
| $R_{lwd}$ (W·m$^{-2}$) | -20.98~20.98 | -2.50~2.50 (-1.63%~1.63%) | -15.54~15.54 (-9.36%~9.36%) |

| | | | |
|---|---|---|---|
| $u$ (m$\cdot$s$^{-1}$) | -1.71~1.71 | -9.47~7.31 (-6.18%~4.77%) | 9.47~-7.31 (5.71%~-4.41%) |
| $T_a$ (K) | -2.08~2.08 | 14.64~-16.94 (9.55%~-11.05%) | -14.64~16.94 (-8.82%~10.20%) |
| SH (kg$\cdot$kg$^{-1}$) | -0.56×10$^{-3}$~0.56×10$^{-3}$ | -0.01~0.01 (-0.01%~0.01%) | 0.01~-0.01 (0.01%~-0.01%) |
| $P_s$ (hPa) | -8.51~8.51 | -0.01~0.01 (-0.01%~0.01%) | 0.01~-0.01 (0.01%~-0.01%) |

The spatial resolution of the final flux products should be determined by the lowest input of the source data, which was mentioned in the revised manuscript (P5, L8-9). Thus, the final surface heat flux product should be 10 km. We have corrected this mistake in the manuscript after the quick review and mentioned this issue in the response letter to the quick reviewer comments.

For the temporal resolution, a linear statistical downscaling method was used to derive the hourly meteorological forcing data based on the original 3-hour forcing data and in situ measurements in this study. The general idea is to establish an empirical relationship between each 3-hour in situ measurement. Then, this relationship is applied to meteorological forcing data (P5, L17-21). For example, $T_{a00}, T_{a01}$ and $T_{a03}$ represent the in situ air temperature measurements from six stations at 00h, 01h and 03h, respectively. Thus $T_{a00} = [a_1, a_2, a_3, a_4, a_5, a_6]$, $T_{a01} = [b_1, b_2, b_3, b_4, b_5, b_6]$, and $T_{a03} = [c_1, c_2, c_3, c_4, c_5, c_6]$. Then, the linear equation $T_{a01} = k_1 T_{a00} + k_2 T_{a03}$ can be solved. According to the meteorological forcing data at 00h and 03h, the plateau scale $T_a$ at 01h can be achieved by the following formula.

$$\begin{pmatrix} b_{11} & \cdots & b_{1n} \\ \vdots & \ddots & \vdots \\ b_{m1} & \cdots & b_{mn} \end{pmatrix} = k_1 \begin{pmatrix} a_{11} & \cdots & a_{1n} \\ \vdots & \ddots & \vdots \\ a_{m1} & \cdots & a_{mn} \end{pmatrix} + k_2 \begin{pmatrix} c_{11} & \cdots & c_{1n} \\ \vdots & \ddots & \vdots \\ c_{m1} & \cdots & c_{mn} \end{pmatrix}$$

where $a$, $b$ and $c$ represent meteorological forcing data at 00h, 01h and 03h respectively; and m and n represent total rows and columns, respectively, of the grid data. The meteorological forcing data at other times can be achieved similarly determined.

*(2) The major supporting for the conclusion of a better performance of the proposed product than the GLDAS produce is based on the comparison with the observational data. The authors use the Bowen ratio calibration method to improve the observed data. We know the validity of the Bowen ratio method varies distinctly in different environments due to the different fulfillment of assumptions. As a result, certain biases will be brought into the*

*observational data, and this can mislead the comparison. First, it is not clear in the text that if the comparison is under the same condition that the observational data are all corrected with the Bowen ratio method. Second, even if using the similar observational data for the comparison, the biases from the correction can still distort the RMSE. Hence, I would suggest directly using the observed data for comparison. Besides, since the data quality of eddy covariance measurements may vary at the 6 stations, comparison on the indicators like RMSE at each station separately may provide more information.*

**Author Response:** It should be noted that the in situ flux data have been flagged by steady state tests and developed conditions tests according to Foken and Wichura (1996) and Foken et al. (2004). Steady conditions mean that all statistical parameters do not vary with time. The flux-variance similarity was used to test the development of turbulent conditions. A data quality of only QA<5 was chosen to make the comparison. Therefore, the comparison is under similar conditions. The above information has been included in the text (P7, L27-28; P8, L1-2).

The Bowen ratio correction method was only used to correct the in situ latent heat flux measurements and was not used for the other three energy balance components (radiation heat flux, sensible heat flux and soil heat flux). The following equation was used to perform the Bowen ratio correction.

$$BRLE = \frac{1}{1+\beta}(R_n - G_0)$$

where $BRLE$ is the latent heat flux after correction and Bowen ratio $\beta = \frac{H}{LE}$. $R_n$ and $G_0$ are net radiation flux and soil heat flux, respectively.

As you mentioned, the validity of the Bowen ratio method varies distinctly in different environments due to the different assumptions. We try to use the original latent heat flux measurements to perform the validation. The following figure shows the comparison between surface latent heat fluxes estimated by the SEBS model with in situ measurements. All corresponding values for the latent heat flux comparison have been replaced in Table 4 (P18). It can be seen that that indicators for latent heat flux have been changed but the they will not influence the general results of this paper. All information on the Bowen ratio correction has been deleted from the original manuscript.

[Figure]

**Figure: Validation of surface latent heat fluxes estimated by the SEBS model with in situ measurements (a. BJ station; b. D105 station; c. Linzhi station; d. MS3478 station; e. Nam Co station; f. QOMS station).**

For your last question, since the data quality of eddy covariance measurements may vary at the 6 stations, a separate comparison of the indicators, such as RMSE, at each station may provide more information. For the data quality, QA<5 was chosen to ensure the flux measurements are under similar steady state and developed conditions; thus, it is not necessary to make a comparison at each station separately. There will be some differences among those stations, but most of these differences can be explained by the quality of the input forcing data, as shown in the sensitivity test.

*(3) The product provided by the authors is produced based on the input data with a spatial resolution no less than 10 km. The authors compare it with a product with a spatial resolution of 25 km. While the scale of the stations normally represents a scale of about less than 1 km. The authors should give some explanation about their comparability.*

**Author Response:** The scale problem you mentioned is an important and difficult problem to be solved in the quantitative remote sensing and atmospheric research field. First, the datasets generated by our methods need to be validated by comparison with the observation dataset. The eddy covariance system is widely accepted as a direct measurement of energy heat fluxes and has been used to validate satellite estimations (Fisher et al, 2008, Ma et al, 2006, Su 2002). Some errors can be caused due to scale mismatch among the stations between the SEBS product and GLDAS product. This issue has been discussed in Section 4.1 (P9, L16-19) and Section 5. Because of the relatively homogeneous land surface conditions of the field stations, this effect should have been minimized in our study. Scintillometry is possibly the most convenient method to measure fluxes at a 1-10 km scale. Unfortunately, this device is lacking over the TP. If we have enough in situ measurements within a grid scale of 10 km or 25 km, an average or weighted average of the measurements can be directly used to reduce some uncertainties caused by scale mismatch. However, for well-known reasons, it is very difficult to carry out such measurements in the TP with the harsh environment and climate conditions. The above discussions have been added to the revised manuscript. (P11, L27-30; P12, L1-4)

**Minor issues:**

*(1) P5-line 13-25: the authors validate the forcing data and find the notable variance. These differences can further propagate to the product. Please discuss its relation to the final product.*

**Author Response:** Yes, we totally agree with you. According to your suggestion, a sensitivity test was carried out to test how the RMSE of forcing data can affect the sensible heat flux and latent heat flux. As shown in Figure 4, three sites located in the northern, western and southeastern part of the TP were randomly selected to perform the sensitivity analysis. All input meteorological forcing parameters in Table 3 ($R_{swd}, R_{lwd}$, u, $T_a$, SH, $P_s$) are selected. The original sensible heat flux and latent heat flux from the SEBS model are used as reference values. The RMSEs of different

forcing data were used as perturbations. As shown in Table 5, the sensible heat flux is highly sensitive to $R_{swd}$, u and $T_a$, while the latent heat flux is very sensitive to $R_{swd}$, $R_{lwd}$ and $T_a$. Both sensible heat flux and latent heat flux are not sensitive to errors of SH and $P_s$. As $R_{swd}$ varies from –68.5 Wm$^{-2}$to 68.5 Wm$^{-2}$, the induced latent heat flux uncertainty ranges from -29.75 Wm$^{-2}$ to 35.86 Wm$^{-2}$. Similarly, the sensible heat flux is very sensitive to $T_a$. When $T_a$has an uncertainty from -2.08 K to 2.08 K, the induced sensible heat flux uncertainty ranges from 14.64 Wm$^{-2}$ to -16.94 Wm$^{-2}$. All the above information has been added to the revised manuscript. (P9, L6-16)

[Figure]

**Figure 4: Locations of the three sites (marked by pentagrams) used to carry out sensitivity tests of the meteorological forcing input data. The legend of the color map indicates the elevation above mean sea level in meters.**

**Table 5: Uncertainties for each meteorological forcing variable and the induced changes in $H_s$ and LE.**

| Variables | Assumed Uncertainty | Induced Uncertainty of $H_s$ | Induced Uncertainty of LE |
|---|---|---|---|
| $R_{swd}$ (W·m$^{-2}$) | -68.50~68.5 | -12.34~6.22 (-8.05%~4.06%) | -29.75~35.86 (-17.92%~21.60%) |
| $R_{lwd}$ (W·m$^{-2}$) | -20.98~20.98 | -2.50~2.50 (-1.63%~1.63%) | -15.54~15.54 (-9.36%~9.36%) |
| u (m·s$^{-1}$) | -1.71~1.71 | -9.47~7.31 (-6.18%~4.77%) | 9.47~-7.31 (5.71%~-4.41%) |
| $T_a$ (K) | -2.08~2.08 | 14.64~-16.94 (9.55%~-11.05%) | -14.64~16.94 (-8.82%~10.20%) |
| SH (kg·kg$^{-1}$) | -0.56×10$^{-3}$~0.56×10$^{-3}$ | -0.01~0.01 (-0.01%~0.01%) | 0.01~-0.01 (0.01%~-0.01%) |
| $P_s$ (hPa) | -8.51~8.51 | -0.01~0.01 (-0.01%~0.01%) | 0.01~-0.01 (0.01%~-0.01%) |

*(2) P8-line 2-6: The introduction of the GLDAS dataset should not belong to Result. The authors should introduce it more in light of its importance for comparison.*

**Author Response:** The introduction of the GLDAS dataset has been moved to section 2 (P5, L26-30). We also introduce the importance of a comparison with the GLDAS product as follows (P5, L22-26).

The high-quality, global land surface fields provided by GLDAS support weather and climate prediction, water resources applications, and water cycle investigations. Since the GLDAS data have been widely used, it is meaningful to compare our satellite estimation with these high-quality data to further prove the accuracy of our estimations.

*(3) P8-line 5: what high accuracy?*

**Author Response:** This term has been replaced with 'high quality'. (P11, L18)

*(4) P9-line 15-27: the authors describe the feature of diurnal variation of hourly flux map. Are there any special in comparison on our general understanding?*

**Author Response:** The qualitative description of diurnal variation in the hourly flux map is aligned with general knowledge. This alignment can further prove the effectiveness of our estimation method and validate the final estimation results. In the revised manuscript, we also added some quantitative expressions to improve the content. (P10)

*(5) Table 4: add values of the same indicators for all sites.*

**Author Response:** For the data quality, QA<5 was chosen to ensure the flux measurements are under similar steady state and developed conditions; thus, it is not necessary to make a comparison at each station separately. Furthermore, if we list the same indicators for all sites, two additional pages will be needed to show this content. Adding this information may make the text difficult to read. Therefore, we would like to keep the original Table 4, but the indicators for latent heat flux have been replaced by the values before the Bowen ratio correction, as you suggested.

*(6) Figure 1: the caption is too brief. The same problems for other plots. What is the right plot?*

**Author Response:** Thank you for this comment. The right plot illustrates the location of the Tibetan Plateau in China. We have improved all figure captions in the revised

manuscript. (P19-24)

*(7) Figure 4: the scale of the axis is misleading. Besides, how do you choose the representative days for each month? Choose the nice one? Please describe what they are in subpanels.*

**Author Response:** Figure 4 (now Fig. 5) has been redrawn to improve its clarity. We also added additional explanations in the figure caption to prevent ambiguity. We did not select the nice days. Instead, the monthly mean value is shown in Figure 5. The subpanels are described in the caption of the new Figure 5. (P23)

**Response to Reviewer #2**

*This is an integral work for estimation of land surface heat fluxes based on remote sensing data, reanalysis meteorological data, and in-situ observations. The derived land surface heat flux, more like a heat flux dataset, was evaluated using observations of six eddy-covariance sites on the Tibetan Plateau (TP). And then, the diurnal and seasonal variations of the heat fluxes were also analyzed. This is of general interest for the readers of this journal. The TP is notorious for its lack of meteorological observations, which cripples the predictive power of numerical models for this region. The land surface heat fluxes are crucial for understanding energy and water cycle and also are the boundary conditions for numerical weather and climate simulations. This paper provides an integral investigation for land surface heat fluxes over the TP which will helps better understanding the land-atmosphere interactions over this region. More importantly, this paper is one of the very few works to estimate land surface heat fluxes over the TP using high temporal resolution geostationary satellite data. The manuscript is well organized. Numbers of work are integrated into this paper, and abundant discussions are presented as well. I suggest acceptance after a minor revision.*

**Author Response:** We would like to thank Reviewer #2 for the positive and constructive comments. All your thoughtful comments and suggestions have been taken into account to improve our manuscript. Please find our point-by-point responses below.

*(1) P1, L16: "which" → "where".*

**Author Response:** This item has been corrected. (P1, L16)

*(2) P1, L18-19: Change the sentence to "However, the high temporal-resolution information about the plateau-scale land surface heat fluxes has lacked for a long time, which significantly limit the understanding of diurnal variations in land-atmosphere interactions."*

**Author Response:** Thank you for this detailed suggestion. The sentence has been revised. (P1, L18-20)

*(3) P1, L20: "a" → "the".*

**Author Response:** This item has been corrected. (P1, L21)

*(4) P1, L21: "with a spatial resolution"→"at a spatial resolution".*
**Author Response:** This item has been corrected. (P1, L22)

*(5) P4, L9: The sentence "These stations are the only stations currently available: : :..." is not accurate. I am quite sure that there are other eddy-covariance sites on the TP apart from the six stations mentioned in the paper.*
**Author Response:** Yes, you are correct. There are several other eddy-covariance sites on the TP. However, these sites belong to different institutes, and the data are not available to the scientific community.

*(6) P7, Equation (11) and (13): "Hs"→"Hs".*
**Author Response:** These items have been corrected. (P7)

*(7) P8, L4: I do not think "Zhong et al., 2011" is a proper reference here. Perhaps you cite the paper which introduces the production of GLDAS data.*
**Author Response:** Thank you for this suggestion. Rodell et al. published a paper in BAMS in 2004 to introduce the GLDAS data. This reference has been added. (P5, L23)

*(8) P8, L12: Provide some references for "traditional polar orbiting satellite" to strengthen your argument.*
**Author Response:** Thank you for your constructive suggestion. The following references have been added. (P8, L23-24)
Ma, Y., Zhong, L., Su, Z., Ishikawa, H., Menenti, M. and Koike, T.: Determination of regional distributions and seasonal variations of land surface heat fluxes from Landsat-7 Ehanced Thematic Mapper data over the central Tibetan Plateau area, J. Geophys. Res.-Atmos., 111, D10305, DOI: 10.1029/2005JD006742, 2006.
Ma W, Ma, Y. and Ishikawa, H.: Evaluation of the SEBS for upscaling the evapotranspiration based on in-situ observations over the Tibetan Plateau, Atmos. Res., 138, 91-97, 2014.

Zou, M., Zhong, L., Ma, Y., Hu, Y., Huang, Z., Xu, K., and Feng, L.: Comparison of two satellite-based evapotranspiration models of the Nagqu River Basin of the Tibetan Plateau. J. Geophys. Res.-Atmos., 123, 3961–3975, DOI: 10.1002/2017JD027965, 2018.

*(9) P10, L23: "land-atmosphere heat flux data" → "land surface heat flux data".*
**Author Response:** This item has been corrected.(P11, L14)

*(10) P10, L24: Delete "using a combination of geostationary and polar orbiting satellite data".*
**Author Response:** This phrase has been deleted.

*(11) The English need substantial improvement. Please find a native speaker to help you to polish the manuscript.*
**Author Response:** According to your suggestion, the revised manuscript has been edited by a native English speaker.

**Response to Reviewer #4**

*High temporal resolution surface heat fluxes are very important for land-atmosphere interactions. In this manuscript, land surface temperature from polar and geostationary satellite are both used and fed into surface energy balance equation. The results are validated with flux tower observations, and finally hourly surface heat fluxes with 5 km spatial resolution are generated over TP based on the developed SEB scheme. Generally, the manuscript is interesting and well written. It can be published with minor revisions.*

**Author Response:** We would like to sincerely thank the reviewer for the thoughtful comments and suggestions. Please see our responses to your comments and suggestions below.

*(1) Page 2, Line 30: I think the authors missed an important kind of method (data assimilation method) for surface heat flux estimations based on remotely sensed LST. Some reference are as follows,*

*Abdolghafoorian, A., Farhadi, L., Bateni, S.M., Margulis, S., Xu, T.R. (2017). Characterizing the effect of vegetation dynamics on the bulk heat transfer coefficient to improve variational estimation of surface turbulent fluxes. J. Hydrometeorol. 18, 321–333.*

*Bateni, S.M., Entekhabi, D., & Castelli, F. (2013), Mapping evaporation and estimation of surface control of evaporation using remotely sensed land surface temperature from a constellation of satellites, Water Resour. Res., 49, 950-968, doi:10.1002/wrcr.20071.*

*Crow, W.T., & Kustas, W.P. (2005). Utility of assimilating surface radiometric temperature observations for evaporative fraction and heat transfer coefficient retrieval, Bound-Lay. Meteorol., 115(1), 105-130, doi:10.1007/s10546-004-2121-0.*

*Xu, T, Bateni, S.M., Liang, S., Entekhabi, D., & Mao, K. (2014). Estimation of surface turbulent heat fluxes via variational assimilation of sequences of land surface temperatures from Geostationary Operational Environmental Satellites, J. Geophys. Res., 119, 10,780-10,798, doi:10.1002/2014JD021814.*

*Xu, T.R., He, X.L., Bateni, S.M., Auligne, T., Liu, S.M., Xu, Z.W., Zhou, J., Mao, K.B.(2019). Mapping Regional Turbulent Heat Fluxes via Variational*

*Assimilation of Land Surface Temperature Data from Polar Orbiting Satellites, Remote Sensing of Environment, 221, 444-461, https://doi.org/10.1016/j.rse.2018.11.023.*

**Author Response:** Thank you for your helpful suggestion. We totally agree with you. Land surface temperature and vegetation information from satellites have been used to estimate regional land surface heat fluxes by different assimilation techniques in recent years. All the above references together with the following comments have been added to the revised manuscript. (P3, L9-15)

In recent years, land surface temperature and vegetation index data retrieved from satellites have been successfully assimilated in the variational data assimilation (VDA) frameworks to estimate surface heat fluxes (Crow and Kustas 2005; Bateni et al., 2013; Xu et al., 2014; Abdolghafoorian et al., 2017; Xu et al., 2019). This kind of method does not require any empirical or site-specific relationships and can provide temporally continuous surface heat flux estimates from discrete spaceborne land surface temperature (LST) observations (Xu et al., 2014).

*(2) How to derive 5 km and hourly surface heat fluxes with 10 km and 3 hour forcing data?*

**Author Response:** The final resolution of our product should be determined by the lowest resolution of the source data. Thus, the final surface heat flux product should be 10 km. We corrected this mistake in the manuscript after the quick review by one of the reviewers.

It should also be noted that the forcing dataset of ITPCAS has a spatial resolution of 10 km and a temporal resolution of 3 hours. For the temporal resolution, a linear statistical downscaling method was used to derive hourly meteorological forcing data based on the original 3-hour forcing data and in situ measurements in this study. The general idea is to establish an empirical relationship between each 3-hour measurement. Then, this relationship is applied to meteorological forcing data (P5, L17-21). For example, $T_{a00}, T_{a01}$ and $T_{a03}$ represent the in situ air temperature measurements from six stations at 00 h, 01 h and 03 h, respectively. Thus, $T_{a00} = [a_1, a_2, a_3, a_4, a_5, a_6]$, $T_{a01} = [b_1, b_2, b_3, b_4, b_5, b_6]$, and $T_{a03} = [c_1, c_2, c_3, c_4, c_5, c_6]$, Then, the linear equation $T_{a01} = k_1 T_{a00} + k_2 T_{a03}$ can be solved. According to the meteorological forcing data at 00h and 03h, the plateau scale $T_a$ at 01h can be

achieved by the following formula.

$$\begin{pmatrix} b_{11} & \cdots & b_{1n} \\ \vdots & \ddots & \vdots \\ b_{m1} & \cdots & b_{mn} \end{pmatrix} = k_1 \begin{pmatrix} a_{11} & \cdots & a_{1n} \\ \vdots & \ddots & \vdots \\ a_{m1} & \cdots & a_{mn} \end{pmatrix} + k_2 \begin{pmatrix} c_{11} & \cdots & c_{1n} \\ \vdots & \ddots & \vdots \\ c_{m1} & \cdots & c_{mn} \end{pmatrix}$$

where $a$, $b$ and $c$ represent meteorological forcing data at 00 h, 01 h and 03 h, respectively; and m and n represent total rows and columns, respectively, of the grid data. The meteorological forcing data at other times can be similarly determined.

*(3) In equation 5, sensible heat flux is represented as Hs, while it is H in equation 11. They should be the same in one manuscript.*

**Author Response:** This item has been corrected to keep the same format. (P7)

*(4) What is the time period of this study? as well as validation results in Table 3.*

**Author Response:** The time period for all meteorological data and satellite data covers the whole year of 2008. This information has been added in section 2. (P5, L9-10)

*(5) Figure2: the 'ITPCAS' is a name of institute, not data. It should be changed into 'Meteorological data' or something else.*

**Author Response:** 'ITPCAS' in Figure 2 has been replaced with 'Meteorological forcing data'. (P20)

*(6) Figure 3: the estimated G0 has a big bias against ground measurements. This is because G0 is parameterized with Rn. G0 and Rn do not have the same diurnal variation shape. The G0 peak values are usually later than Rn. However, the parameterization did not consider this. The authors may discuss this in the manuscript.*

**Author Response:** Thank you for this insightful comment. We discuss the large bias in estimated soil heat flux as follows. (P8, L7-12)
It should be noted that some bias exists between the estimated soil heat flux and ground measurements because soil heat flux is parameterized with net radiation flux (equation (8)). However, soil heat flux and net radiation flux do not have the same diurnal variation shape. The soil heat flux peak values are usually later than the net radiation flux peak values, which was not taken into account in the parameterization.

Thus, development of a better parameterization scheme for soil heat flux is needed.

*(7) Figure 4: usually, the observations were drawn by open cycles, and estimations are drawn by solid lines.*

**Author Response:** Figure 4 (now Figure 5) has been redrawn according to your suggestion. (P23)

*(8) Why Rn is underestimated from June to Aug. at BJ site in figure 4? Why H (LE) is underestimated (overestimated) from Jan. to May? The authors should give some explanations.*

**Author Response:** As shown by the surface radiation balance equation (equation (6)), the downward short radiation is the main incoming energy. A comparison was made between the forcing data and in situ downward radiation at BJ station. From June to August, the monthly diurnal MB was -4.87 $Wm^{-2}$, which explains why the derived net radiation flux was underestimated by the SEBS model from June to August. This phenomenon was also found in the study by Yang et al. (2010). As for the time period from January to May, the underestimation of sensible heat flux was mainly caused by the negative bias of the land-atmosphere air temperature difference. The MB for the land-atmosphere difference could be -5.69 K from January to May. As there is a complementary relationship between sensible heat flux and latent heat flux, the corresponding latent heat flux tends to be overestimated. This discussion has been added to the revised manuscript. (P10, L8-18)

*(9) Figure 5: the authors give two days of diurnal cycles over TP. The results are from which day and which year? It should be noted on figure 5. In addition, why you choose these two days?*

**Author Response:** It should be noted here that the diurnal cycles of land surface heat flux are based on the annual mean of 2008. The top panels are sensible heat flux, and the bottom panels are latent heat flux. We have added this information in the figure caption. (P24)

[revised manuscript text omitted]
. ~~The 'energy imbalance' is an important research issue and has been widely reported in former studies (Twine et al., 2000; Wilson et al., 2002; Wolf et al., 2008; Majozi et al., 2017; Pan et al., 2017). If these measurements were not corrected and directly used to compare with estimates, some discrepancies would appear. Therefore, a so-called Bowen ratio correction method (Twine et al., 2000; Wilson et al., 2002; Hu et al, 2018) is used to process the in situ flux measurements. The results show that the energy closure ratio can be improved by approximately 20% for different stations (Hu et al, 2018). Then, the corrected in situ flux measurements are used to validate the satellite estimates.~~ As shown in Fig. 3a, 3b, 3c and 3d, the estimates of surface energy budget components show reasonable

agreement with the in situ measurements. The RMSEs for the net radiation flux, sensible heat flux, latent heat flux and soil heat flux are 76.63 Wm$^{-2}$, 60.29 Wm$^{-2}$, 71.03 Wm$^{-2}$ and 37.5 Wm$^{-2}$, respectively. The total validation numbers (N) are more than 3837 to make the results much more representative and convincing. It should be noted that some bias exists between the estimated soil heat flux and ground measurements. This is because soil heat flux is parameterized with net radiation flux (Eequation (8)). However, soil heat flux and net radiation flux do not have the same diurnal variation shape. The soil heat flux peak values are usually later than the net radiation flux peak values, which was not taken into account in the. The parameterization did not take this into account. Thus, development of a How to better parameterization scheme fore the soil heat flux remains an open issue 
[revised manuscript text omitted]

Lee, X., and Hu, X.: Forest air fluxes of carbon, water and energy over non-flat terrain. Bound. Lay. Meteorol., 103(2), 277-301, DOI: 10.1023/A:1014508928693, 2002

Liu, X., and Chen, B.: Climatic warming in the Tibetan Plateau during recent decades. Int. J. Climatol., 20(14), 1729-1742, 2000.

Ma W, Ma, Y. and Ishikawa, H.: Evaluation of the SEBS for upscaling the evapotranspiration based on in-situ observations over the Tibetan Plateau, Atmos. Res., 138, 91-97, 2014.

Ma, W., Ma, Y., Li, M., Hu, Z., Zhong, L., Su, Z., Ishikawa, H., and Wang, J.: Estimating surface fluxes over the north Tibetan Plateau area with ASTER imagery. Hydrol. Earth Syst. Sc., 13(1), 57-67, DOI: 10.5194/hess-13-57-2009, 2009.

Ma, Y., Ma, W., Zhong, L., Hu, Z., Li, M., Zhu, Z., Han, C., Wang, B., and Liu, X.: Monitoring and

modeling the Tibetan Plateau's climate system and its impact on east Asia. Sci. Rep., 7, 44574 , DOI: 10.1038/srep44574, 2017.

Ma, Y., Menenti, M., Feddes, R. A., and Wang, J.: The analysis of the land surface heterogeneity and its impact on atmospheric variables and the aerodynamic and thermodynamic roughness lengths. J. Geophys. Res.-Atmos., 113, D08113, DOI: 10.1029/2007JD009124, 2008.

Ma, Y., Su, Z., Koike, T., Yao, T., Ishikawa, H., Ueno, K. I., and Menenti, M.: On measuring and remote sensing surface energy partitioning over the Tibetan Plateau—from GAME/Tibet to CAMP/Tibet. Phys. Chem. Earth, 28(1-3), 63-74, DOI: 10.1016/S1474-7065(03)00008-1, 2003.

Ma, Y., Su, Z., Li, Z., Koike, T., and Menenti, M.: Determination of regional net radiation and soil heat flux over a heterogeneous landscape of the Tibetan Plateau. Hydrol. Process., 16(15), 2963-2971, DOI: 10.1002/hyp.1079, 2002.

Ma, Y., Zhong, L., Su, Z., Ishikawa, H., Menenti, M., and Koike, T.: Determination of regional distributions and seasonal variations of land surface heat fluxes from Landsat-7 Ehanced Thematic Mapper data over the central Tibetan Plateau area. J. Geophys. Res.-Atmos., 111, D10305, DOI: 10.1029/2005JD006742, 2006.

Majozi, N. P., Mannaerts, C. M., Ramoelo, A., Mathieu, R. S., Nickless, A., and Verhoef, W.: Analysing surface energy balance closure and partitioning over a semi-arid savanna FLUXNET site in Skukuza, Kruger National Park, South Africa. Hydrol. Earth Syst. Sc., 21(7), 3401-3415, DOI: 10.5194/hess-21-3401-2017, 2017.

Mauder, M., Oncley, S. P., Vogt, R., Weidinger, T., Ribeiro, L., Bernhofer, C., Foken, T., Kohsiek, W., De, Bruin HAR., and Liu, H.: The energy balance experiment EBEX-2000. Part II: Intercomparison of eddy covariance sensors and post field data processing methods. Bound. Lay. Meteorol., 123(1), 29-54, DOI: 10.1007/s10546-006-9139-4, 2007.

Norman, J., Kustas, W. P., and Humes, K. S.: Source approach for estimating soil and vegetation energy fluxes in observations of directional radiometric surface temperature. Agr. Forest Meteorol, 77(3-4), 263-293, DOI: 10.1016/0168-1923(95)02265-Y, 1995.

Oku, Y., and Ishikawa, H.: Estimation of land surface temperature over the Tibetan Plateau using GMS data. J. Appl. Meteorol., 43, 548-561, 2004.

Oku, Y., Ishikawa, H., Haginoya, S., and Su, Z.: Estimation of land surface heat fluxes over the Tibetan Plateau using GMS data. J. Appl. Meteorol. Clim., 46, 183-195, 2007.

Pan, X., Liu, Y., Fan, X., and Gan, G.: Two energy balance closure approaches: applications and comparisons over an oasis-desert ecotone. J. Arid Land, 9(1), 51-64, DOI: 10.1007/s40333-016-0063-2, 2017.

Qiu, J.: Monsoon Melee. Science, 340(6139), 1400-1401, DOI: 10.1126/science.340.6139.1400, 2013.

Rodell, M., Houser, P. R., Jambor, U., Gottschalck, J., Mitchell, K., Meng, C. J., Arsenault, K., Cosgrove, B., Radakovich, J., Bosilovich, M., Entin, J. K., Walker, J. P., Lohmann, D., and Toll, D.: The Global Land Data Assimilation System. B. Am Meteorol. Soc., 85(3), 381-394, DOI: 10.1175/BAMS-85-3-381, 2004.

Roerink, G. J., Su, Z., and Menenti, M.: S-SEBI: A simple remote sensing algorithm to estimate the surface energy balance. Phys. Chem. Earth, Part B: Hydrology, Oceans and Atmosphere, 25(2), 147-157, DOI: 10.1016/S1464-1909(99)00128-8, 2000.

Sánchez, J. M., Scavone, G., Caselles, V., Valor, E., Copertino, V. A., and Telesca, V.: Monitoring daily evapotranspiration at a regional scale from Landsat-TM and ETM+ data: Application to the Basilicata region. J. Hydrol., 351(1-2), 58-70, DOI: 10.1016/j.jhydrol.2007.11.041, 2008.

Seneviratne, S. I., and Stöckli, R.: The role of land-atmosphere interactions for climate variability in Europe [C]. Climate Variability and Extremes During the Past 100 years, Switzerland, Springer, 179-193, 2008.

Su, Z.: The Surface Energy Balance System (SEBS) for estimation of turbulent heat fluxes. Hydrol. Earth Syst. Sc., 6(1), 85-100, DOI: 10.5194/hess-6-85-2002, 2002.

Twine, T. E., Kustas, W. P., Norman, J. M., Cook, D. R., Houser, P., Meyers, T. P., Prueger, J. H., Starks, P. J., and Wesely, M. L.: Correcting eddy covariance flux underestimates over a grassland. Agr. Forest Meteorol., 103(3), 279-300, DOI: 10.1016/S0168-1923(00)00123-4, 2000.

Wilson, K., Goldstein, A., Falge, E., Aubinet, M., Baldocchi, D., Berbigier, P., Bernhofer, C., Ceulemans, R., Dolman, H., Field, C., Grelle, A., Ibrom, A., Law, B. E., Kowalski, A., Meyers, T., Moncrieff, J., Monson, R., Oechel, W., Tenhunen, J., Valentini, R., and Verma, S.: Energy balance closure at FLUXNET sites. Agr. Forest Meteorol., 113(1), 223-243, DOI: 10.1016/S0168-1923(02)00109-0, 2002.

Wolf, A., Saliendra, N., Akshalov, K., Johnson, D. A., and Laca, E.: Effects of different eddy covariance correction schemes on energy balance closure and comparisons with the modified Bowen ratio system. Agr. Forest Meteorol., 148(6-7), 942-952, DOI:

10.1016/j.agrformet.2008.01.005, 2008.

[revised manuscript text omitted]